# Land in Water: The Study of Land Reclamation and Artificial Islands Formation in the UAE Coastal Zone: A Remote Sensing and GIS Perspective

P. Subraelu [1,*], Abdel Azim Ebraheem [1], Mohsen Sherif [1,2], Ahmed Sefelnasr [1], M. M. Yagoub [3] and Kakani Nageswara Rao [4]

1   National Water and Energy Center, United Arab Emirates University, Al Ain P.O. Box 15551, United Arab Emirates
2   Civil and Environmental Engineering Department, College of Engineering, United Arab Emirates University, Al Ain P.O. Box 15551, United Arab Emirates
3   Department of Geography, College of Humanities and Social Sciences, United Arab Emirates University, Al Ain P.O. Box 15551, United Arab Emirates
4   Department of Geo-Engineering, Andhra University, Andhra Pradesh, Visakhapatnam 530003, India
*   Correspondence: subraelupakam@gmail.com

**Abstract:** The United Arab Emirate's rapid population growth is coupled with an increase in the consumption of natural resources such as fresh air, sunlight, land, and water. In the past two decades, the demand for land has augmented both away from the coast and significantly near the coast. Within coastal zones, artificial reclamation of land in the sea is the most desirable way to meet the demand for land necessary for the development of the most modern urban areas. Seaward reclamation (land in the water) necessitates the construction of artificially reclaimed areas that are extended into the sea using innovative modern construction techniques. The majority of these building requirements are necessitated by a number of key factors and have diverse outcomes. Even though this type of urban expansion is not new, the scale and motivations of land reclamation have been drastically altered due to geological and human-induced factors. The purpose of this paper is to assess the increase in seaward land expansion, particularly in the seven UAE coastal emirates. Using satellite data, particularly from 1990 to 2021, the total increase in land due to newly developed reclaimed areas in all UAE coastal emirates is calculated. Satellite images from the Landsat series are used to analyze the tremendous growth since the early 2000s. In addition, the study of shoreline maps of 1990, 2000, 2010, and 2021 for the seven emirates revealed that the 22 km long Ajman and UAQ front coast experienced a notable shoreline retreat with a net erosion area of 300 m$^2$ and an annual rate of 30 my$^{-1}$ over the past 21 years (2000–2021). Depending on the type of construction design used to describe the process, a methodical sorting is also recommended. The impacts of the Dubai offshore reclaimed islands on the adjacent coastlines in Ajman and Umm Al Quwain (UAQ), as well as the potential impact of earthquake tremors along the Zagros fold belt region, are the subjects of this study. In this study, all seven coastal emirates are considered, and the largest reclamation projects are located in Dubai, Abu Dhabi, Ras-Al Khaimah (RAK), and Fujairah, with Dubai leading the way; it has expanded its coastal areas by more than 68 km$^2$ at present, and another 35 km$^2$ will be reclaimed soon to finish Palm Deira.

**Keywords:** coastal emirates; artificial reclamation of islands; geoengineered shorelines; land use/land cover; UAE shoreline; geospatial analysis; GIS; remote sensing

## 1. Introduction

The most important global indicator of population growth is likely the extensive and quick expansion of urban sprawl [1]. Megacities have emerged as a result of growing urban built-up areas and rising population, particularly in coastal regions of the world. Increasing

population and related high levels of built-up expansion, especially in coastal towns, have contributed to the tense situation brought about by the modernization of infrastructure [2,3]. Regarding this, land reclamation (land in water) has begun and created new land from the ocean. The simplest method entails filling the designated area with enormous quantities of rock, followed by cement, clay, and soil until the desired height is reached [4]. After the Rotterdam port in the Netherlands was modified in the 1970s [5–7], the first significant land reclamation in a megacity began. Land reclamation in the sea has necessitated significant geotechnical procedures that have artificially reclaimed land by modifying the surface morphology along and past the coastlines in coastal cities. The exceptional examples include, to name a few, the various Palm resorts of Dubai, Durrat Al Bahrain, Amwaj Island, Bahrain, Pearl Island, and Ras Laffan, Qatar, the new Hong Kong airport, and the development of "Baia deLuanda" in Luanda, Angola. Other nations, including Japan [8], Korea [9,10], Indonesia [11], and Lima (Peru), have previously reclaimed land to extend it into the ocean. Construction of a new Mumbai, the second-highest populated coastal city in the world, is another example [12]. In the last three to four decades, reclamation projects in China have substantially altered the coastlines [13–15].

The construction of human-made structures in the oceans has the potential to considerably alter the Earth's surface, thereby disrupting geomorphic processes such as coastal land subsidence, hydrology, coastal erosion, and longshore/nearshore sediment dynamics [16]. The Food and Agriculture Organization's (FAO) land use/land cover classification system [17] classified and described artificial surfaces and associated areas as in the urban category. Building entirely on newly reclaimed land in the ocean requires sophisticated geoengineering techniques that are not required when constructing new structures on the existing land surface. The construction of artificially reclaimed land is entirely driven by anthropogenic practices operating at various stages. Multiple processes at the international and national level are driven by geopolitics and economic stability. Land reclamation projects have been initiated on a national scale as a result of rapid financial development and subsequent urban sprawl expansion [18]. Unconditional population growth and concurrent urbanization expansion are essential aspects of global environmental processes [19]. Construction of land along coastlines and in the water, for example, in Dubai [19–21], Bahrain [22,23], Qatar [21,24], and Hong Kong [25], has greatly aided in the expansion of the real estate industry; however, questions have been raised about its environmental impact and long-term durability [23,26].

Sudden and persistent growth in global urban sprawl has aided in the development of coastal cities, which in turn has placed a tremendous burden on environmental and sociodemographic systems [27]. To accommodate the ever-increasing population and their needs, the global built-up areas grew. This is most prevalent in cities in Asia and the Middle East, such as Shanghai [28], Hong Kong [29], Manila [30], Jakarta [31], Dubai [32], Abu Dhabi [33], Bahrain [34], Doha, and Mumbai [35], where growth has occurred at a rapid rate. Currently, thirty million people reside in these coastal urbanized cities, which are only 200 km from the coastline. This number is anticipated to double within the next three years [12,36]. Urban settlements of this type paved the way for monetary benefits to coastal cities, helped to improve transport facilities, yielded high capital gains from tourism improvement, and enhanced trade and commerce; however, this type of development also posed environmental concerns. Sustaining a balance between current economic growth, societal development, and environmental protection is difficult. In the coming years, the demand for absolute built-up area in terms of vacant physical spaces will grow to be of critical importance. It is anticipated that megacities will arise along the coasts in the future. It becomes increasingly important to align with the landward and maritime expansion pattern. Due to their geographic location and impact on the coastal, marine, and regional environments, coastal megacities are the primary agents of environmental transformation [37].

To comprehend the changes in the coastline caused by human activities, it is essential to keep track of the former land reclamation areas. Examining reclaimed areas has

numerous benefits for the positive development of any coastal city. The past information on dredging and reclamation activities is of great assistance to the coastal managers and higher authorities of land reclamation projects; this will allow them to construct profitable developments. Satellite image information enables high-quality observations and high temporal resolutions. According to [38–40], there has been a significant increase in coastal land reclamation, and the trend of artificially reclaimed land has been upward for the past 50 years. Using remote sensing data, contemporary research in this field has quantified coastal changes, particularly land reclamation, in recent years. Furthermore, [41,42] evaluated the negative effects of financial development, high population growth, and urban sprawl rate on coastal developments and concluded that a prosperous and wealthy economy is the primary reason for these kinds of complex coastal projects. Optical satellite images, such as the Landsat time series from 1976 to 2015, are used to identify Ningbo's shoreline changes [43] and the island's coastline changes using Landsat images from 1984 to 2009 [44]. Various kinds of satellite image data are utilized to determine the morphodynamics of the coast [45,46]. Additionally, Sen Gupta [47] used the Google Earth Engine (GGE) platform, remote sensing images, and the Joint Research Center Global Surface Water dataset to assess the trend of coastal land reclamation in China's three main deltaic regions. Fung created a map illustrating the growth of Hong Kong from 1979 to 1987 using data from Landsat MSS and SPOT High Resolution Visible (HRV) [48]. In Egypt, satellite images are used to study the artificial land reclamation [49].

Due to global warming, eustatic sea levels are rising, primarily as a result of thermal expansion of seawater and the addition of water from ice melt [50–53]. The IPCC (2019) [54] projected an increase of 0.61 m to 1.10 m by 2100 [55], and sea levels will persist to rise after 2100, but the extent and rate of the rise will depend on the future consumption of hydrocarbons [56]. More studies have predicted that sea levels will rise by approximately 7.5 m over the end of next century [57]. The sea level rise (SLR) will have a direct effect on the various land cover features along the coast, which are meant to be resourceful and thickly populated, but are also low elevation and therefore vulnerable to severe erosion, which threatens natural resources and expensive infrastructure [58–60]. In addition to the SLR, the local land subsidence is deemed important to comprehend, as it has been reported that it is caused by widespread groundwater, oil, and gas extraction [61–64] and sediments present there are naturally compacted due to their own weight [65,66]. Due to their low elevation and dense economic and sociocultural exposure, artificially reclaimed coastal areas are highly susceptible to eustatic SLR and spring tide-induced storm surges. During extreme weather conditions, such as storm surges, sea levels will increase in height and frequency and intensity of flooding [67–69]. There will apparently be a dramatic increase in the intensity of tropical cyclones, resulting in extreme storm surges [70,71]. Clearly, infrastructure development in coastal zones becomes important as the effects of coastal flooding evolve [72]. As unprecedented urban sprawl is anticipated to increase in the majority of the world's coastal cities [73], it is anticipated that an overall increase will occur and classification of their susceptibility to coastal flooding will be conducted in the future. In contrast, neither future development nor current reclamation projects account for the threat of extreme coastal flooding [74]. Finally, it is essential to examine the impact of coastal flooding on reclaimed lands and their socioeconomic vulnerability, which will facilitate the proper planning and execution, thereby enabling the construction of coastal flood-resistant cities.

Now referring to the Kingdoms of the Arab Peninsula, it is common knowledge that rapid economic growth and cultural affluence allowed these nations to transform from remote desert provinces into superpowers with rapidly expanding coastlines [75]. On the other hand, as a result of the population explosion of the last thirty to forty years and the resulting urbanization, there has been an extraordinary increase in the energy and water needs of these nations, which has put a significant strain on their land resources [76–80]. For the purpose of resolving this issue, marine resources have garnered significant interest. Consequently, Gulf countries are creating artificial lands in the waters of the Arabian Gulf

to address the rapidly growing urban sprawl and population and, as a result, gaining industrial, financial, and decisive advantages [81,82]. On the reclaimed land areas in the region, tremendously high-quality communities are being constructed. The United Arab Emirates (UAE), Qatar, and Bahrain are investing large sums in these enormous construction projects. Saudi Arabia is also extending land into the sea on Tarut Island for urban expansion. UAE is one of the wealthiest nations per capita in the world [83], with a rapidly growing population along the coastline [84] among all the countries in the Arabian Gulf region. In addition, Dubai is at the forefront of five new massive offshore island communities that are being developed as part of world-famous projects. They are referred to as "The Palm Jebel Ali," "The Palm Jumeirah," "The World Islands," "The Dubai Waterfront," and "The Palm Deira," respectively. On reclaimed land in Doha, a similar project known as "The Pearl Qatar" has developed. In the Kingdom of Bahrain, "Diyar Al Muharraq," "Durrat Al Bahrain," "Dilmunia Island," and "Northern City" were created on reclaimed land. These islands are designed and constructed to increase the length of their coastlines, allowing for the construction of numerous beachfront mansions and resorts. In addition, tens of thousands of residential, commercial, and recreational complexes are constructed in these regions. These developments include high-rise apartment complexes.

The majority of previous research on coastal land reclamation has been conducted at the global level but not at the regional or national level [85]. The purpose of this paper is to study the varying spatial extent of coastal land reclamation in UAE at the level of individual emirates and in relation to changing population and urbanization.

This study's primary objective is to evaluate the magnitude of the anthropogenic seaward land development in coastal emirates of the United Arab Emirates from 2000 to 2022, while highlighting the reasons for the extensive expansion of this phenomenal construction in the Emirate of Dubai and its probable potential impacts in recent years. The objectives of this study are as follows:

- To generate coastline maps of the seven coastal emirates from 1990, 2000, and 2010 and correlate them to the situations in 2021.
- To compute the total area developed in each of the coastal emirates during the period of analysis.
- Analyze geomorphic designs created by various geotechnical techniques for the purpose of creating surplus coastal land.
- Consider briefly the possible causes of Dubai's extensive construction and their environmental impacts on land expansion.

## 2. Study Area

The United Arab Emirates (UAE) is located on the Arabian Peninsula, which is a part of the Middle East and shares the Arabian Gulf's coastline. UAE has a total land area of 83,000 km$^2$ and shares maritime borders with the Islamic Republic of Iran to the north and Qatar to the west. The majority of UAE's land area consists of desert and mountains. It shares land borders with Oman in the east, Saudi Arabia in the south and west. The United Arab Emirates (UAE) consists of seven coastal emirates along the Arabian Gulf: Abu Dhabi (Capital), Dubai, Sharjah, Ajman, Umm Al Quwain (UAQ), Ras Al Khaimah (RAK), and Fujairah. The UAE is located between latitudes 22°30′ and 26°10′ north and longitudes 51° and 56°25′ east. On the southern side of the Arabian Gulf, the UAE's coastline stretches for more than one thousand kilometers. The present study region consists of the 550 km long UAE coastline (illustrated by the red box in Figure 1), which encompasses all the coastal Emirates (Figure 1). This study focuses on a 140 km stretch of Abu Dhabi's 520 km coastline because the western portion of the emirate is underdeveloped and there are no land reclamation projects in the western coastal region. Particularly across Fujairah and Ras-Al Khaimah, the northern coastal region of the United Arab Emirates is distinguished by a headland bay configuration with a number of cliffs protruding into the sea, as well as having coastlines that are 90 km and 64 km long, respectively. Meanwhile, Dubai, having a native shoreline length of 72 km, has mostly a sandy and low dune coast. The current

study also concentrates on the UAQ and Ajman coasts, whose shorelines measure 24 km and 16 km, respectively, shown in black boxes (Figure 1). In contrast to Ajman, which has solely a sandy coast, UAQ Emirate mostly has a natural sandy shoreline across the city and a barrier island coast. Abu Dhabi Ports and Khalifa Port in Abu Dhabi, Jebel Ali Port and Rashid Port in Dubai, and Fujairah Port in Fujairah are among the most important ports and trading centers in the UAE. To evaluate coastal land reclamation in the UAE, it is most important to examine population data for each emirate in depth. Dubai has the largest population in the United Arab Emirates, according to a report from Dubai Statistics Center "www.dsc.gov.ae (accessed on 18 August 2022)". In the latter half of the 20th century, the Emirate of Dubai experienced significant expansion and transformation due to its oil and gas industry. The city's thriving economy and desire to expand have led to some of the world's greatest aspiring construction projects, including the world's tallest skyscraper, Burj Khalifa, as well as several shopping centers, amusement parks, hotels, and primarily artificially reclaimed islands (ARIs).

The Arabian Gulf is a vast, flat, warm, semi-enclosed body of water with a complex oceanography and geomorphology [86]. The Arabian Gulf is an approximately 221,000 km² inland epicontinental sea with a length of 816 km, a breadth of 250–300 km, and a maximum width of 370 km. The Arabian Gulf opens into the Gulf of Oman through the 56 km wide strait of Hormuz, which separates the Musandam Peninsula in Saudi Arabia and Qeshm Island in Iran. The Arabian Gulf has a maximum depth of approximately 100 m and an average depth of approximately 50 m. In the Arabian side offshore zone, the depth does not exceed 50 m [87]. A zone with a maximum depth of roughly 100 m is located very close to the Iranian border. The confluence of the Euphrates and Tigris Rivers, which forms the Shatt al-Arab River, forms a delta at the northernmost point of the Arabian Gulf. Other smaller streams from the Iranian Zagros Mountains flow into the Arabian Gulf, forming estuaries. Currently, no streams flow from the Arabian Peninsula into the Arabian Gulf; however, numerous smaller streams and rivers flowed into the Gulf during the Cenozoic period [88]. The Arabian Gulf is located on the continental shelf, with a bottom topography that gradually slopes eastward. No shelf edge exists. Extreme aridity, hot summer temperatures (approximately 50 °C), and the limited separation of the Gulf from the Arabian Sea result in extremely high salinities throughout the basin, as well as the formation of sabkhas and salt precipitation in the coastal lagoons. The peninsula of Qatar on the western edge of the Gulf primarily affects marine currents and sedimentation patterns on the southeast side of the Gulf, i.e., along the UAE coastline. The predominant local winds are referred to as "Shamals," and they blow from the northwest to the southeast, affecting coastal habitats with waves and surface currents [89].

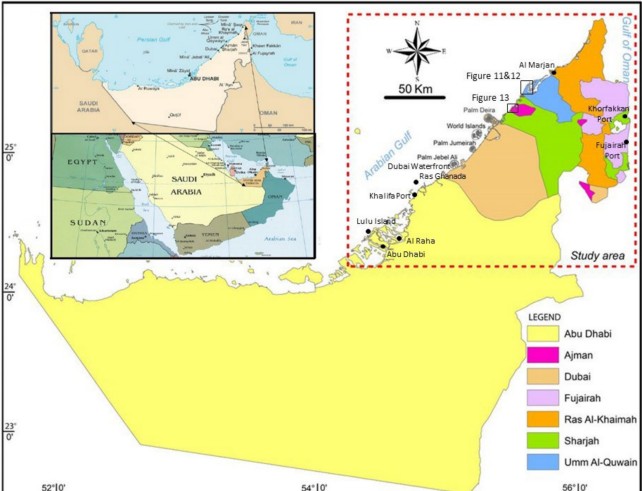

**Figure 1.** Location map of the study area showing the important reclaimed areas in the UAE.

## 3. Materials and Methods

Population data were used as the main criterion for selecting the coastal emirates that would be the subject of a detailed study in order to evaluate the coastal artificial reclamation of land. Due to their coverage, quality, and accessibility, satellite images of the Earth's surface are often used in geospatial research. This research utilized satellite images from the Earth Resource Observation System, specifically Landsat and Sentinel-2 images (Table 1). Landsat images from various time periods were gathered and analyzed to shed light on Dubai's coastal seaward development, shoreline alterations, and land use/land cover. Landsat data from 1990, 2000, and 2010 and Landsat 7 for the most recent year 2021 were used. For each selected coastal emirate, the absolute area was computed using the administrative shoreline boundary from the old images and superimposed on the boundary of the current image on the Landsat images. In addition, Band 7 (Landsat TM) and Band 5 (Landsat 7) were used to digitize the coastline due to their capacity to differentiate between land and ocean. The change in area between the two images was then determined. The most recent satellite image was then evaluated to establish a classification of geographical patterns resulting from the seaward spread of land. All Sentinel-2 images for the year 2022 were precisely georeferenced, enhanced, and smoothly combined into a single mosaic image of Dubai Emirate. The image mosaic is subsequently subset with the aid of Dubai's boundary layer. Similarly, a land use/land cover map was created using the 2022 Sentinnel-2 image in order to study the built-up patterns and the rationale for Dubai Emirate having the most ARIs. Additionally, Dubai's year-wise land reclamation has been established with the aid of temporal Landsat images.

**Table 1.** Data utilized for various parameters.

| Data Type | Data Used |
| --- | --- |
| LULC | Sentinel-2A (Jan 2022) 20 m, spatial resolution |
| Environment protected areas map of Dubai | Dubai Municipality (Environment Department) |
| Shoreline changes | Landsat 5 TM (Aug 1990) 30 m, spatial resolution Landsat 5 TM (Nov 2000) 30 m, Spatial resolution Landsat 7 ETM (Sep 2010) 30 m, Spatial resolution Landsat 7 ETM (Nov 2021) 30 m, Spatial resolution |
| Bathymetric contour maps | Navionics marine charts |
| Population data | Dubai Statistics Center "www.dsc.gov.ae (accessed on 18 August 2022)" Abu Dhabi Statistics Center "www.scad.ae (accessed on 18 August 2022)" |
| Earthquake data | "https://earthquake.usgs.gov/earthquakes/map/ (accessed on 27 July 2020)" |

## 4. Results

### 4.1. Mapping "Building beyond Land in Arabian Gulf Waters Pertaining to the UAE"

Thorough interpretation of the satellite images enabled visualization and calculation of the spatial distribution of land beyond the sea in each of the coastal emirates of the UAE. According to Figure 2, between the 2000s and 2021 there was a total addition of 119 km$^2$ of land reclaimed in the six coastal emirates. Regarding the total area of each emirate, even though Abu Dhabi has an area of 67,327 km$^2$, which is the largest of all the emirates, it has reclaimed only 35 km$^2$, whereas Dubai has a meagre area of 3885 km$^2$ and has the highest population in the UAE (Figure 3), but it has reclaimed 68 km$^2$ which is more than 70% of the total reclaimed area in the UAE and there is more 35 km$^2$ yet to be reclaimed in the

future, in the form of the beautiful Palm Deira and Waterfront islands. Dubai is followed by Abu Dhabi (35 km$^2$), RAK (6.97 km$^2$), and Fujairah (7.58 km$^2$). There are notable differences in reclamation rates in the UAE. For instance, Dubai, with a population now nearing 4 million, has added 68 km$^2$ of land since early 2000 (Figure 4), whereas Abu Dhabi has reclaimed an area of 35 km$^2$ from the late 1980s to 2021 despite having a population of 2.7 million. Coastal reclamation of land has been very rapid in the coastal emirates of the UAE; examples include Abu Dhabi (Figure 5a–c); Dubai (Figure 5d–h); RAK (Figure 5i); and Fujairah (Figure 5j). The ongoing land reclamation project at Palm Deira requires approximately another 25 km$^2$ area to be reclaimed which is between the 15 m and 20 m contour. The ports of Abu Dhabi, Dubai, and Fujairah have expanded seaward mainly through the creation of artificial land (Figure 5c,f,j). RAK and Fujairah also have extended their coastlines with an area of 6.97 km$^2$ and 7.58 km$^2$, respectively.

Since the foundation of the United Arab Emirates (UAE) in 1973, the nation has welcomed huge numbers of expatriates, who have drastically altered the country's landscape. The demand for these people coincided with the economic expansion caused by the enormous oil earnings. Prior to 1975, the demographic status of Dubai was minor. During this time period, the bulk of the community consisted of citizens who were mostly involved in fishing and commerce. In the 45 years since the current economic expansion, the population has increased by more than one thousand percent (Figure 6). In 1975, there were 183,000 residents in Dubai, a number that climbed to more than 2 million in 2015 and reached 3.5 million in 2022. This expansion ranks Dubai's population as among the fastest growing in the world, based on its annual growth rate. In a similar context, foreigners have contributed greatly to this population rise, as they account for almost 88% of the entire population.

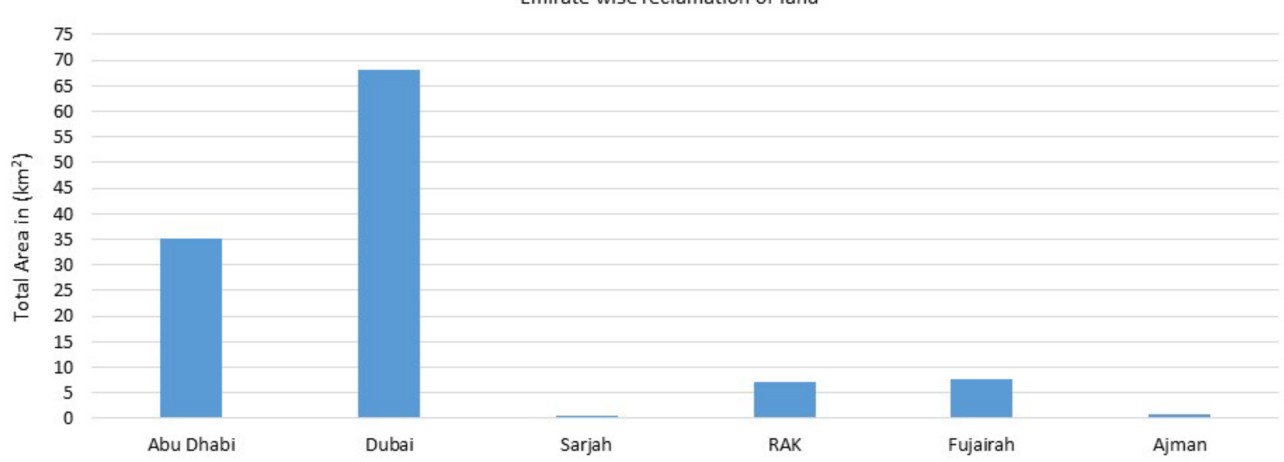

**Figure 2.** Bar graph representing the reclaimed area in each emirate.

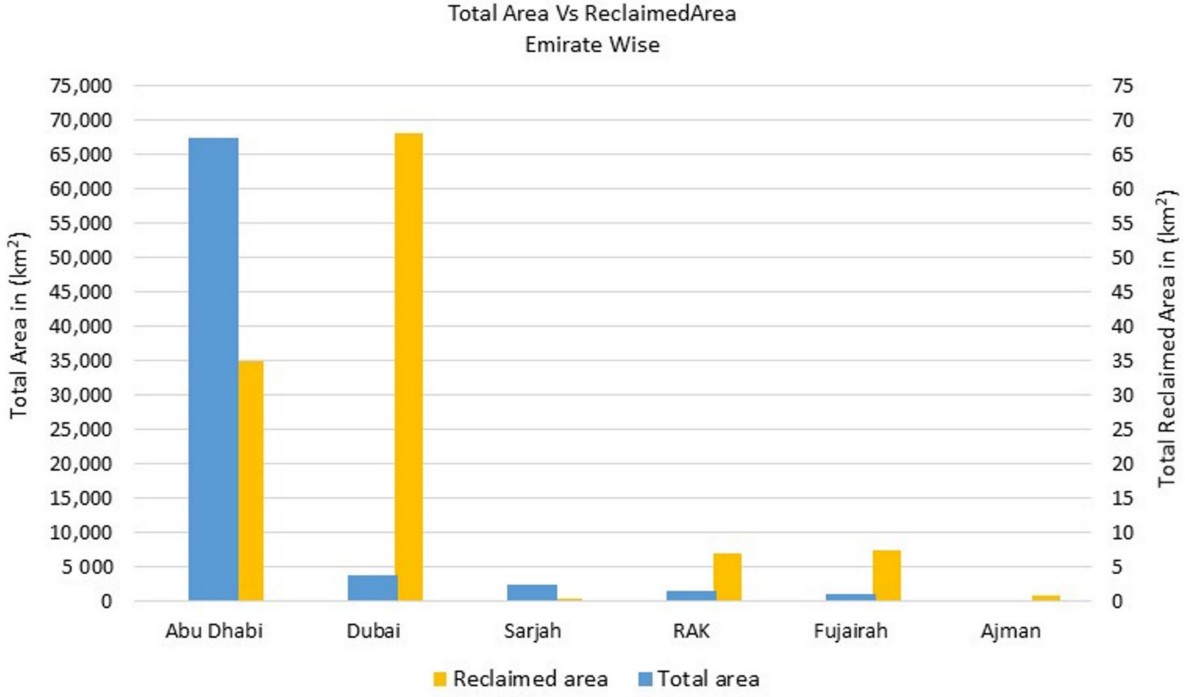

**Figure 3.** Bar graph indicating total area of each emirate vs. reclaimed area of each emirate.

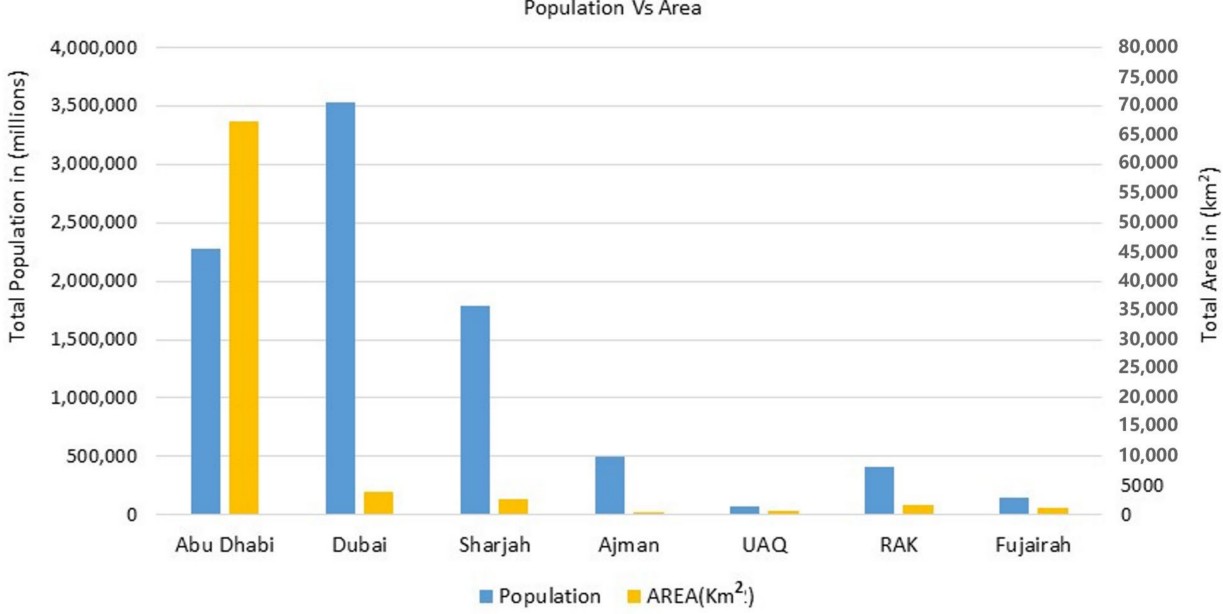

**Figure 4.** Bar graph indicating total population of each emirate vs. total area of each emirate.

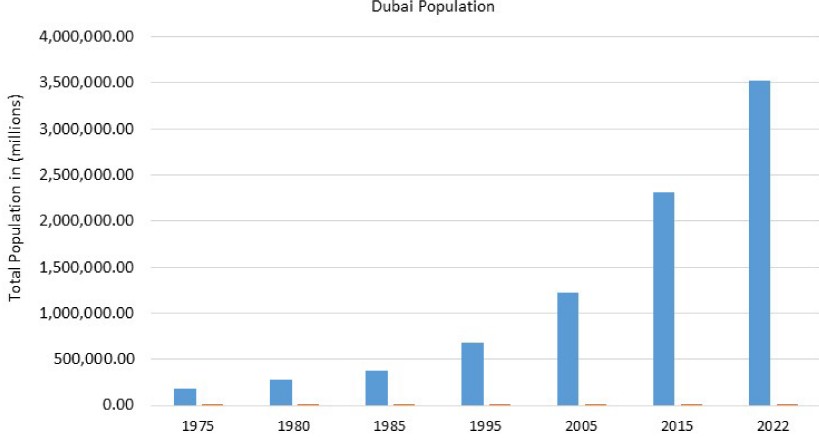

**Figure 5.** Artificial reclamation of land in Abu Dhabi (**a**–**c**); Dubai (**d**–**h**); RAK (**i**); Fujairah (**j**).

**Figure 6.** Bar graph showing the population of Dubai from 1975–2022.

*4.2. Classification of Different Geoengineered Reclaimed Lands in the UAE*

The relationship between the natural and anthropogenic drivers at various levels eventually leads to alteration of the Earth's surface. The interconnection between waves, underwater currents, changing sea levels, storms, and artificial reclaimed islands' responses is complex and increases more when people try to manipulate the coastal processes [90]. Therefore, the physical environment obviously influences the spatial patterns of artificial land construction in the sea, signifying that the development of a particular typology could be a useful means of understanding the process. During the process of construction, the high turbidity levels brought on by the construction activities are one of the key problems encountered during the development of the artificial reclamation of islands. Recent advancements such as ecoengineering, an emerging area that merges engineering and ecological approaches to provide sustainable urban constructions, have been made to minimize the turbidity levels. For instance, Bahrain used geotubes when developing the Amwaj Islands to reduce the effects of dredging. Instead of building a quarry rock-retention dike, developers used geotubes to combat the threat of high turbidity. Due to dredging and reclamation works, this unquestionably succeeded in limiting turbidity [21]. Additionally, using geotubes drastically lowers the carbon emissions linked to construction activities [91]. Turbidity was intended to be reduced to its lowest level using woven polypropylene fabric silt curtains that were 200 m long and 3 m high [92]. This is a fairly new, revolutionary technique that is reasonably priced and has many positive environmental effects. The geotube's real operation entails filling with materials, dewatering by capturing 99% of particles, and finally disposing of solid materials in the project site [93]. Ecoengineering also contributed to the creation of sustainable urban environments in Malaysia. This concept is better suited for regions where there is sufficient space between the coast and artificial islands [94] to encourage the establishment of natural ecosystems including reefs, seagrass beds, mangroves, and saltmarshes to mitigate the effects of powerful waves [95]. When building artificial islands in Hong Kong, a cutting-edge construction technique was applied. Two islands with a combined area of 100,000 m$^2$ were constructed using a deep-insert steel cylinder piling with auxiliary cells method in 6 months as opposed to 3 years [21]. Additionally, this approach reduced the amount of dredging required for the construction of the islands. It finally lessened the project's impact on the maritime environment by further lowering sedimentation and turbidity levels.

Coming to the UAE, satellite images with better temporal resolution assist in detection and monitoring and provide a platform to perceive and distinguish between specific patterns of construction. The different types of construction along the coastline of the six emirates are arranged into two unique geomorphic models on the basis of their physical appearance (Table 2). The categories are segregated as follows: (1) Reclaiming land, which involves extension of artificial land from the already existing shoreline. (2) Offshore construction in the form of new land is carried out, leaving open water between the shoreline and the artificial land. The selected coastal emirates can be classified on the basis of these distinctive types of construction.

**Table 2.** Classification of six coastal emirates based on geoengineering construction patterns.

| Type of Construction Model | Name of the Reclaimed Area |
| --- | --- |
| Expanded land construction | (a) Blue Water Island (Dubai), Figure 5e |
| | (b) Jumeirah Open Beach Islands (Dubai), Figure 5f |
| | (c) Dubai Maritime City (Dubai), Figure 5f |
| | (d) Jebel Ali container terminal (Dubai), Figure 5d |
| | (e) Al Marjan (RAK), Figure 5i |
| | (f) Fujairah Port (Fujairah), Figure 5j |
| | (g) Al Kasir (Abu Dhabi), Figure 5a |
| | (h) Khalifa Port (Abu Dhabi), Figure 5c |
| | (i) Al Raha (Abu Dhabi), Figure 5b |
| Offshore construction | (a) World Islands (Dubai), Figure 5g |
| | (b) Palm Jebel Ali (Dubai), Figure 5d |
| | (c) Palm Jumeirah (Dubai), Figure 5e |
| | (d) Palm Deira (Dubai), Figure 5h |
| | (e) Daria Island (Dubai), Figure 5f |
| | (f) LuLu Island (Abu Dhabi), Figure 5a |

*4.3. Land Use/Land Cover Features of Emirate of Dubai*

As depicted in (Figure 6), Dubai's population has expanded dramatically from 183,187 people in 1975 to 2.381 million in 2015 and to 3.58 million in 2022, at an annual growth rate of about 6%, despite having a minimum constrained surface area of 3885 km$^2$. This denotes a confined area with an annual population increase of more than 10 percent between 1985 and 2021. Dubai reached yearly growth rates of 13.03% between 2003 and 2005, making it one of the world's fastest-growing megacities. In reality, Dubai's demographic rise is among the world's highest [96]. Typically, population expansion induces alterations in land use and land cover trends [97]. This presents an evident challenge to the management and government officials responsible for urban planning, particularly in light of global climate change [98]. In light of this, a complete land use/land cover map of Dubai (Figure 7) was created in order to gain a better understanding of how this massive population is accommodated in such a limited land area.

Dubai's land use/land cover (LULC) characteristics have been evaluated and mapped via on-screen digitization. All land use classes up to and including Level III of the most commonly used National Remote Sensing Center (NRSC) categorization are digitized to the level that they may be interpreted from relatively high-resolution satellite images and substantial field observations. Built-up residential, industrial, commercial, recreation, underdeveloped, plantation, barren scrubland, mudflats, mangroves, wetlands, environmental protected areas, ports, and airports are the LULC classes assessed for Dubai (Figure 7). In all of these land use classes, built-up residential occupies 763 km$^2$, followed by 194 km$^2$ of underdeveloped area and 151 km$^2$ of industrial areas. However, there are seven environmental protected areas in the Emirate of Dubai, namely Al Marmoom Desert conservation area covering 993 km$^2$, Dubai Desert conservation reserve area (DDCR) covering 226 km$^2$, Jebel Ali marine conservation area (JAMC) covering 76.68 km$^2$, Al Wohoosh Dhofar conservation area with 15.05 km$^2$, Ras Al Khor conservation area spreading over 10.13 km$^2$, and Nazwa Mountain and Al Ghaf conservation area covering an area of 1.189 km$^2$.

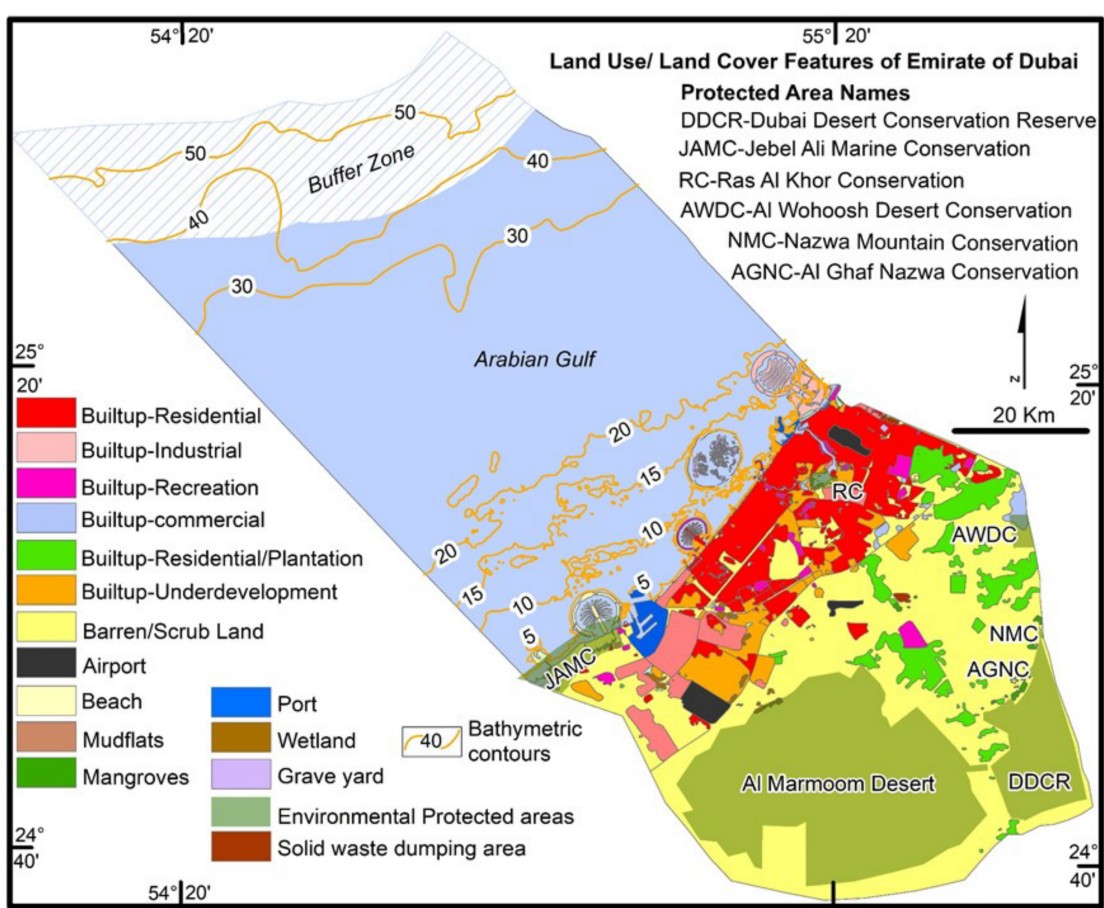

**Figure 7.** A comprehensive land use/land cover map of Dubai.

After reviewing the LULC map of the Emirate of Dubai, it was determined that the combined conservation zones comprise around 1322 km$^2$, or approximately 34% of the entire area including conservation sites; hence, no commercial or residential development is permitted within these boundaries. The government in Dubai is extremely concerned with preserving conservation sites, and they are always a high priority. Therefore, only 2563 km$^2$, or 66% of the area, remains to be developed to accommodate Dubai's rapidly expanding population. Dubai's government initiated the concept of artificial reclamation of islands in 1994, the first in the region, in order to meet the countless needs of metropolitan centers. These islands are largely used for housing, commercial, industrial, trade, tourism, and strategic reasons to control the inexorable growth in population. Moreover, according to [99], using the coastal vulnerability index (CVI) analysis, the whole of the Dubai coastline is categorized as low to moderately vulnerable due to eustatic sea level rise.

## 5. Discussion

### 5.1. Driving Forces behind the Extensive Construction of Land beyond Sea in Dubai

It is vital to study how this type of metropolitan expansion and agglomeration has been so pervasive and swift along the coast. Major towns play a crucial role in extended economical development, and coastal areas in particular have significantly accelerated spatial and financial growth due to a number of causes. Coastal regions are topographically favorable since they offer ample flat surfaces that are more easily accessible than hilly ones. Human intervention at the seaside has increased in the form of the construction of harbors, ports, airports, hotels, resorts, and dwellings since technical improvement is quite obvious and expanding in many regions of the world. This has caused the coastal emirates of the United Arab Emirates, particularly Dubai, to become a renowned illustration of how a certain combination of geographic environments fosters socioeconomic growth. In

addition, there are a number of additional elements, with demographic increase, financial development, institutional action, and social involvement being the most prominent, that should be regarded as potential drivers of seaward land reclamation in Dubai.

The Emirate of Dubai leads the UAE coastal emirates with 68 km$^2$ of reclaimed land, followed by Abu Dhabi with 35 km$^2$, RAK with 6.97 km$^2$, and Fujairah with 7.58 km$^2$. This appears crucial to understanding why this type of the urban growth along the coast has become so pervasive and quick in Dubai. Dubai functions as a driver for the UAE's accelerated economic development. In this setting, various factors have contributed to Dubai's fast spatial and socioeconomic growth. To comprehend the reasons, we must know the history of Dubai and the UAE.

Dubai is a model of a rapidly developing megacity. In 1957, Dubai Municipality was established, and the initial urban plan was devised, resulting in the creation of a road network and a new town center with the construction of contemporary concrete block structures [32]. Since 1975 through to the present, the city expansion has spread tremendously from the old city center (along the creek) in all directions, especially as a result of the Dubai government investing more money in constructing harbors, ports, industrial and residential districts, as well as other commercial centers. The city's quick growth converted it from a tiny regional economic center into a global commerce hub. During the past two decades, Dubai's urban area has grown in all directions, the economic growth has quickened, investments have increased, and major projects have been planned to promote urbanization. The overall urban area rose from a mere around 53 km$^2$ in 1975 [32] to over 1200 km$^2$ in 2021, due to Dubai's growth (about 2500% in 45 years). This significant rate makes Dubai one of the world's most rapidly expanding cities.

The first artificial reclamation of land in Dubai, southwest of Dubai Creek, began in 1994 with the construction of the world's only seven-star hotel, Burj Al Arab, which reclaimed approximately 0.049 km$^2$ of land and was finished in 1999 (Figure 8). With the building of Palm Jebel Ali, Palm Jumeirah, World Islands, and a portion of Palm Deira, the real rate of land reclamation accelerated over the next ten years, i.e., from 2001 to 2010. During this time, an amazing 60,489 km$^2$ has been added to the inventory of reclaimed territories along the Dubai coastline. All of these spectacular initiatives will increase Dubai's coastline by numerous kilometers. In addition, the subsequent five years (2010–2015) saw the construction of five new reclaimed islands, including Dubai Waterfront, which is to the southwest of Palm Jebel Ali, and Jebel Ali container terminal, which is to the north of Palm Jebel Ali, Blue Waters Island, which is to the south of Palm Jumeirah, Daria Island, which is to the north of Dubai Canal, and Pearl Jumeirah, which is to the south of Dubai Maritime City. These five islands have contributed 5. 08 km$^2$ to the total area, bringing the total to 65,569 km$^2$. The entire amount of reclaimed area offshore of Dubai is 67,349.9 km$^2$. As Palm Deira located in the north of Dubai's coastal metropolis has yet to be completed, approximately 24.7 km$^2$ of land must be reclaimed during the next year in order to finish Palm Deira (Figure 8).

In addition, financial drivers play a significant part in propelling the seaward extension of territory through the building of artificially reclaimed coastal land. For example, [13] Tian et al., (2016) found a favorable correlation between China's GDP and land reclamation, particularly from 1985–2010. Moussavi and Aghaei (2013) [100] also revealed substantial relationships between economic policies, growth, technical progress, and urban expansion in Dubai in the form of real estate developments. The growth of Mumbai is likewise closely proportional to its financial growth [35], despite the fact that colonial policies are responsible for the reclamation of land that created a megacity [101]. In our fast-paced world, trade and business are crucial to connecting the megacities. Utilizing the appropriate technology is crucial to the development of the accompanying infrastructure. This entails an increase in transportation infrastructure, primarily in coastal communities, and the development of waterfront infrastructure to facilitate the flow of products and services. Coastal emirates such as Abu Dhabi, Dubai, Sharjah, Ajman, Fujairah, and Ras Al Khaimah have begun extensive seaward land extension to meet rising local, regional, and

international demand. Instances throughout the entire Gulf demonstrated that financial strategies and technical development are the driving forces behind coastal land modification and the creation of artificial reclaimed territory. The Palm resorts in Dubai [100], Amwaj Island in Bahrain [92], and the Pearl development in Doha are three notable examples in the region of sophisticated coastal dredging equipment. This quick progress of coastal land reclamation is also greatly helped by the technology revolution, which has led to significant geoengineering of the shoreline [4,90]. Increasing harbor and port capacity in response to a rise in trade and commerce appears to be a prevalent reason for land extension along the coast. Again, this is due to economic factors, as the rising demand for goods and services necessitates the expansion of these terminals' physical capacity. Satellite images reveal that Abu Dhabi, Dubai, and Fujairah have all engaged in extensive land modification and construction in order to incorporate transit infrastructure in order to sustain their continued growth. Keeping in mind that the production of new artificial land represents a significant financial barrier, freshly developed residential plots are sometimes designed only for the economically wealthy who can afford these places. This is evident in the emirates of Dubai, Abu Dhabi, and RAK, where massive expenditures have been made in pursuit of economic gains. In this regard, the ever-increasing demand for new urban reclaimed land for diverse uses, such as transportation and settlement, has prompted the establishment of appropriate urban planning laws and legislation [32,33]. Typically, seaward urban land extension (land in the sea) is a direct indication of population and financial growth, and the development of artificially reclaimed land is viewed as an investment to sustain and boost economic growth [102].

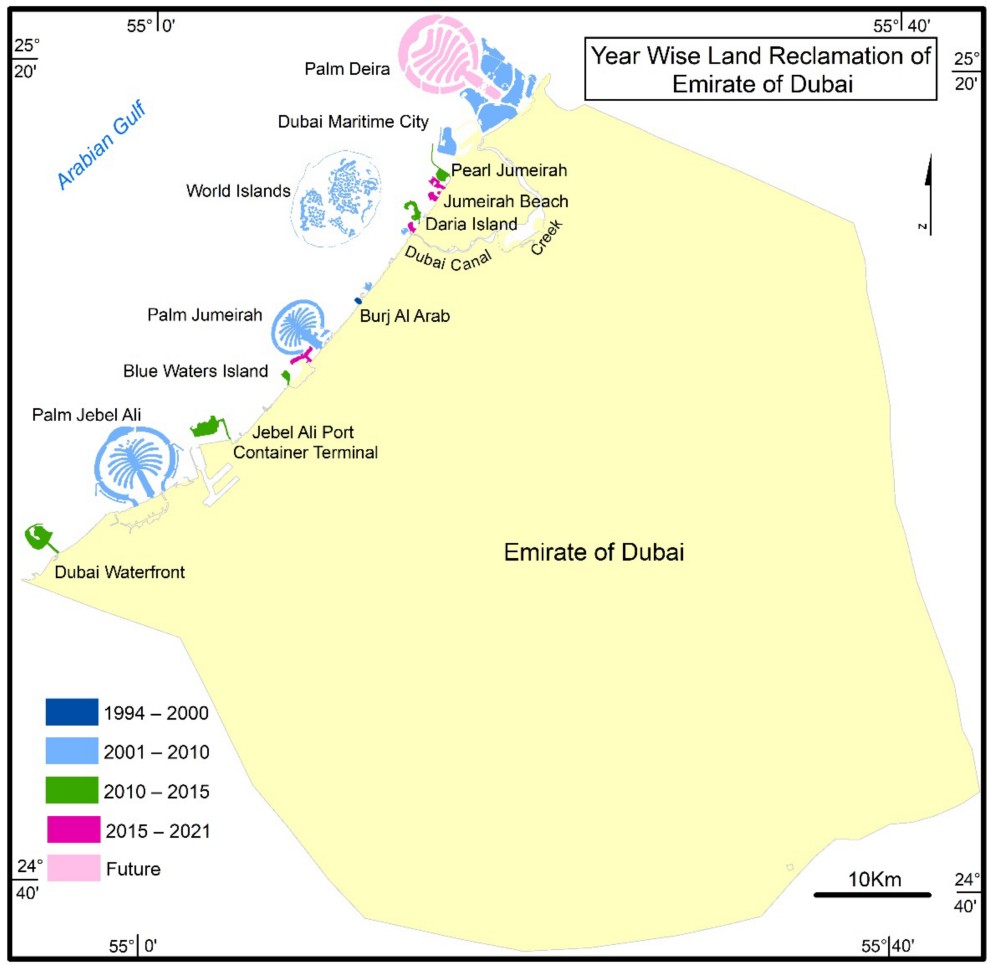

**Figure 8.** Year-wise land reclamation of Dubai.

*5.2. Possible Potential Impacts and Methods to Mitigate the Risks of Artificial Reclamation of Land*

It is generally acknowledged that the artificial reclamation of coastal land raises numerous environmental difficulties [23,103–106]. Such effects are not usually considered, and similar expansions are sometimes deemed to be extremely controversial. The reclamation of land in the shape of an island chain in the South China Sea, whereby material was scraped from the ocean floor, has harmed coral reefs and other marine ecosystem components [107]. It has been extensively documented [23,108,109] that the creation of artificial land has caused a decline of biodiversity in the coastal regions of Bahrain and China. Due to the discharge of construction debris and wastewater during the process of creating artificial land, Hong Kong's marine water has become very polluted [110]. Other ecological hazards resulting from the offshore mining of the building material have been documented [111–113].

In the UAE, many international, regional, and law-making institutions oversee the development of the marine areas and artificial islands in the Arabian Gulf. Therefore, strong policies, consolidation of existing legislation, employing modern methods, workforce development, continuous monitoring, and strict implementation, accompanied by advanced engineering, are deemed necessary to achieve the goal of sustainable growth in the UAE [114,115]. During the development of Palm Jumeirah in Dubai and Khalifa Port in Abu Dhabi, officials relocated corals to safer regions (https://gulfnews.com/uae/environment/corals-from-palm-jebel-ali-to-be-relocated-to-jumeirah-1.701755 and https://www.adports.ae/abu-dhabi-ports-unveils-khalifa-port-coral-relocation-plan) (Accessed on 27 August 2022). Detailed guidelines were developed in 2007 by the Abu Dhabi Urban Planning Council (UPC), and these comprehensive regulations are highly effective when fully implemented [84]. However, strengthening these legislative agendas by the governments in all gigantic constructions sprouting all along coast of the UAE will aid in reducing coastal biodiversity depletion.

5.2.1. Potential Impacts and Risks of ARI Pertaining to the UAE due to Earthquakes Both within the UAE and Iran

As evidenced by occurrences in Haiti, Japan, India, and the United States, earthquakes are a serious disaster that necessitates heightened vigilance and prudent preparation. We must be aware of life-saving responses and preparedness and measures applicable to both small-scale and catastrophic earthquakes. The means of determining, evaluating, and lowering vulnerability to hazards will significantly contribute to the reduction of societal vulnerability [116–119]. Here, preparedness includes the actions that possess the ability to save lives, reduce property destruction, and enhance public control over the response to a disaster [120].

Even though the Arabian Peninsula is regarded as a safer region, the main urban centers in the UAE, such as Dubai, Abu Dhabi, Sharjah, RAK, UAQ, and Fujairah, are vulnerable to the dangers of seismic activity [121]. During the past 35 years, however, the United Arab Emirates has experienced seismic events ranging from minor to moderate magnitude, with a peak value of 5.1 Mw occurring within the Masafi area (northeastern part of the UAE on 11 March 2002) and causing tremors and shaking buildings in major UAE cities [122]. Due to the presence of more artificially reclaimed islands, the magnitude of the tremors in the cities of the UAE, despite being small, appears to become an important topic for research and evaluation. As "low to moderate risk is not synonymous with no risk," Al-Amiri [123,124] identifies 25 seismogenic origin zones in the Arab Peninsula and surrounding nations. They include rift systems, strike slip and fault zones, joints, lineaments, and subduction and collision zones (Figure 9). The Zagros Fold marking the plate boundary between the Arabian and Eurasian tectonic plates is a highly seismically active region [125]. This thrust frequently experiences moderate to strong ($M_w > 5$) earthquakes. In the last 11 years, the USGS (https://earthquake.usgs.gov/earthquakes/map/) (Accessed on 15 September 2022) has recorded approximately 1545 earthquakes in this region (80–400 km away from Dubai and

NE Emirates), and the major earthquakes with a magnitude greater than 5.5 in the last year are listed below in Table 3. Moreover, in the past year, there have been approximately 245 earthquakes, of which four were of high magnitude ($\geq$6), causing residents of Dubai and Sharjah to flee their homes (https://www.khaleejtimes.com/emergencies/uae-residents-report-strong-tremors-after-multiple-earthquakes-kill-at-least-five-in-iran) (Accessed on 27 July 2022), 19 were of moderate magnitude ($\geq$5), and the remaining 227 were of low magnitude ($\geq$4). These earthquakes can induce devastating damage to the high-rise structures of the United Arab Emirates (UAE) on the mainland and also on the reclaimed islands, particularly to the towering, non-earthquake-resilient building structures that pose a high degree of risk from distanced earthquakes; consequently, it is evident that damage in the UAE can be expected as a result of the large earthquakes in southern Iran [126,127].

The UAE has taken numerous practical actions towards prevention and mitigation, most notably the increased density of the National Seismic Networks, the formation of the National Emergency and Crisis Management Authority (NCEMA) in 2007, and the modification of earthquake-related building standards [128] (The National 2013). The government's proactive approach to earthquake catastrophe demonstrates an increasing understanding of the problem's gravity. In addition, reclaimed land is conceivably more vulnerable to liquefaction and slope failure [20,129], so the threat of these earthquakes will be greater on reclaimed islands and in their building complexes. The majority of the reclaimed material used in these regions is dredged carbonate sand, and little is known about the fracture zones nearby. This carbonate sand can cause compaction that results in deformations [130], thereby threatening the solidity of ground structures and reclaimed human-made infrastructure [131]. All of the aforementioned factors make building complexes, particularly high-rise buildings constructed on reclaimed islands, quite hazardous and susceptible to seismic damage. The process of liquefaction causes ground compaction [132], thereby endangering the solidity of ground construction. Therefore, continuous monitoring of deformations is essential for reclaimed land areas in the UAE. As documented in [133], a region of Kuwait experienced liquefaction as a result of previous earthquakes in the region.

**Table 3.** The major earthquakes with magnitude more than 5.5 in the region for the last year.

| S. No | Date | Origin Time (UTC) | Location Lat/Long | Magnitude $M_w$ | Distance from UAE (km) | Region |
|-------|------|-------------------|-------------------|-----------------|------------------------|--------|
| 1 | 2021-11-14 | 12:08:38.812 | 27.7266/56.0716 | 6.4 | 200 | Zagros Fold belt area |
| 2 | 2022-07-01 | 23:25:13.465 | 26.8878/55.3213 | 6 | 150 | Zagros Fold belt area |
| 3 | 2022-07-01 | 21:32:07.977 | 26.9061/55.2389 | 6 | 160 | Zagros Fold belt area |
| 4 | 2021-11-14 | 12:07:03.595 | 27.7158/56.0743 | 6 | 200 | Zagros Fold belt area |
| 5 | 2022-03-16 | 23:15:47.266 | 27.0219/54.5843 | 5.9 | 200 | Zagros Fold belt area |
| 6 | 2022-07-01 | 23:24:13.113 | 26.8843/55.2572 | 5.7 | 150 | Zagros Fold belt area |
| 7 | 2022-07-23 | 16:09:07.209 | 26.9956/55.3766 | 5.6 | 140 | Zagros Fold belt area |
| 8 | 2022-06-25 | 03:37:14.329 | 26.7341/54.2683 | 5.6 | 200 | Zagros Fold belt area |
| 9 | 2022-06-15 | 06:06:02.539 | 26.6789/54.212 | 5.5 | 200 | Zagros Fold belt area |

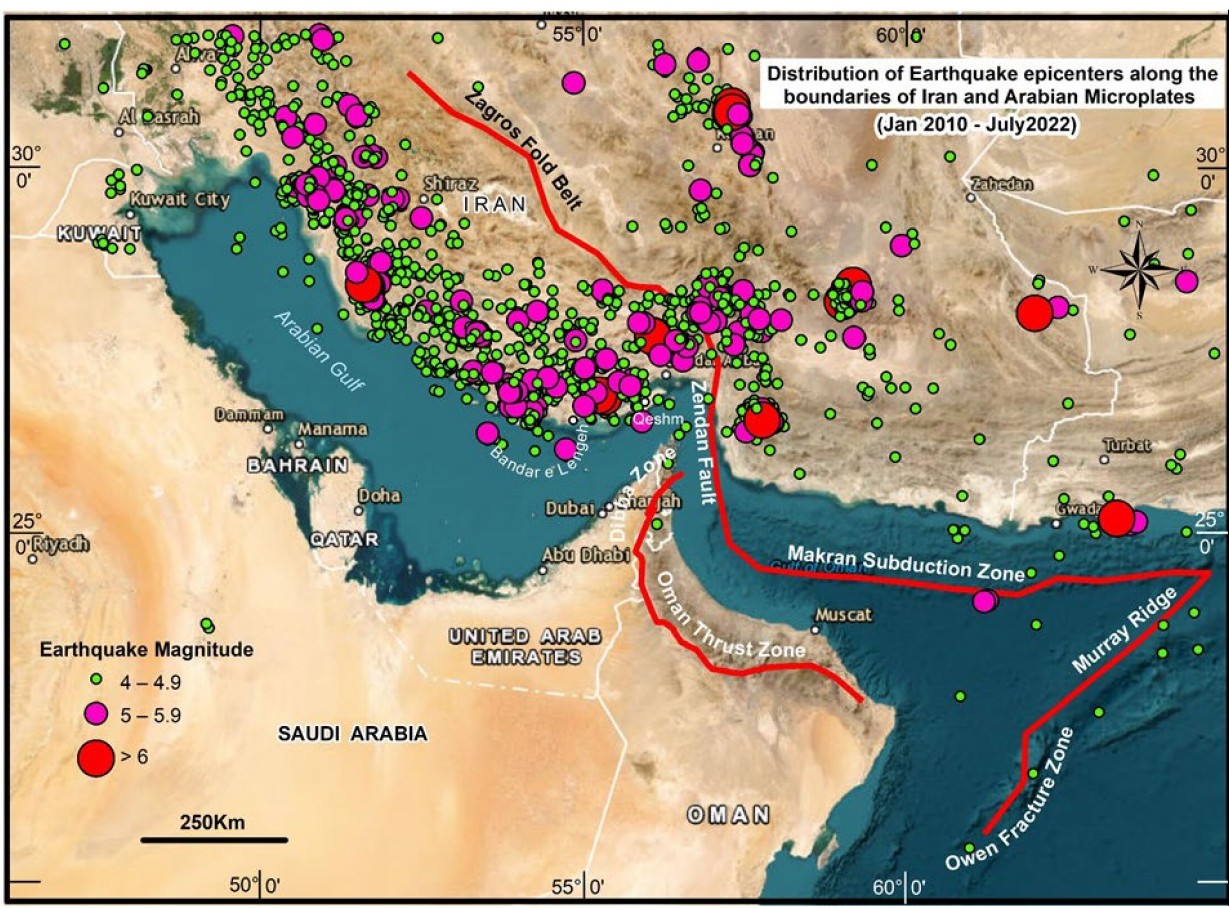

**Figure 9.** Distribution of earthquake epicenters along the boundaries of Iranian and Arabian Microplates (compiled from George Pararas-Carayannis, 2006, [134]; Jamal A. Abdalla and Azm S. Al-Homoud, 2004, [135]).

### 5.2.2. Potential Impacts and Risks of ARIs in the Type of Shoreline Retreat across the Different Emirates of UAE

All of the Arabian Gulf's coastal zones are resourceful and densely populated, but they are also extremely low-lying and therefore highly susceptible to a variety of coastal hazards. Beaches are the most common landforms along coastal zones, and they undergo frequent changes in shape and size as a result of seasonal shifts or sporadic conditions of greater wave energy during stormy weather. In reality, beaches are dynamic landforms that should be considered part of the sea and not the land. Once anthropogenic activities permanently occupy temporary beaches, complications will arise. Beaches are the most valuable asset of coastal regions because they provide beautiful recreational opportunities for tourists and locals. In addition, beaches cushion the impact of high sea conditions, protecting the coastal land and people behind them. We should also understand that the rivers that drain into the ocean, eroding cliffs and associated coastal features, and sometimes the offshore zone, are the sources of beach material. Waves, tides, and currents in the ocean are continuously at work on these materials, helping to transport them along the shoreline and depositing them at specific locations along the coast where conditions are favorable for deposition. The beaches erode during severe storms to accommodate the increased volume of water and wave force. The beaches that develop within bays or between two breakwaters are more dynamic than those that develop along the straight coasts. In the majority of beaches, seasonal changes in the longshore drift cause changes in the beach configuration, with beach material undergoing erosion in the lee of an updrift headland or breakwater and deposition near the breakwater's down-coast location.

Umm Al Quwain (UAQ) and Ajman Coast—A Probable Victim of Construction of Artificial Reclaimed Lands

As these emirates are located in a straight line, the case of the UAQ and Ajman coast likely serves as an example of how the construction of palm islands and other artificially reclaimed islands in Dubai could result in pronounced beach erosion. With these palm constructions, the equilibrium of the shoreline along the UAQ and Ajman coasts has been altered, particularly near the Ajman creek and UAQ front coast, which have shifted to the leeward side of the reclaimed islands that are being constructed in the Arabian Gulf. Erosion occurred along the UAQ and Ajman coastlines. In order to validate the field's erosion, it is necessary to collect field data. The shorelines of Ajman and UAQ have undergone significant spatial and temporal transformations over the past three decades, as revealed by a comparison of four sets of satellite images. Between the 1990s and 2000, the region initially experienced more deposition than erosion. The UAQ and Ajman coastlines gained approximately 295 m$^2$ and experienced 100 m$^2$ of net erosion. In the last two decades, erosion has dominated over accretion in the region, reversing the previous trend. A comparison of (Landsat) image data from the years 2000 and 2020 exposed a net loss of approximately 300 m$^2$ of area across the region and deposition of 130 m$^2$.

Overall, shoreline deposition was greater during the timeframe 1990–2000, whilst erosion exceeded deposition during the successive 20-year period between 2000 and 2021. During the period 1990–2000, the average rate of accretion was 29.5 my$^{-1}$, with the majority of deposition occurring along the UAQ coast. In contrast, during the period 2000–2021, the rate of erosion (land loss) is 30 my$^{-1}$, predominantly along the Ajman and UAQ front coasts (Figure 10). In addition, recent research has uncovered exposed villa compound wall basements along the UAQ coast, besides a barrier spit erosion [56]. During 2021 and 2022, our field observations in the region revealed a severely damaged concrete structure (Figure 11a) and a collapsed compound wall (Figure 11b) on the northern UAQ coast. There was increased coastal erosion to the south of the UAQ coast, as evidenced by the exposed extensive rock surface and exposed underground pipeline (Figure 12b), and the sea waters directly impinging on the villa (Figure 12c). Further south towards Ajman, a restaurant is completely in the intertidal zone (Figure 13a), the basement of a concrete building was eroded by the high wave conditions (Figure 13b), and a concrete structure was completely destroyed by the strong waves (Figure 13c). The total length of different sections of the UAQ coast where deposition predominated between 1990 and 2000 was 12 km, whereas the remaining two km were subjected to erosion (Figure 10a). Notably, 85% of UAQ's front coast experienced accretion, while 15% of the coast was eroded during this time frame. However, during the successive 20 years between 2000 and 2021, erosion occurred along 10 km of the total 14 km while deposition occurred along the remaining four km (Figure 10b). Seventy-one percent (10 km) of the UAQ front coast experienced erosion, while 29% (4 km) experienced deposition. Due to the extensive erosion observed along these coasts, the local government planned a number of shipwrecks along the UAQ and Ajman coasts to limit the erosion by acting as temporary breakwaters and controlling the ongoing erosion by trapping longshore sediment and depositing it on the coast (Figures 11c and 12a,d).

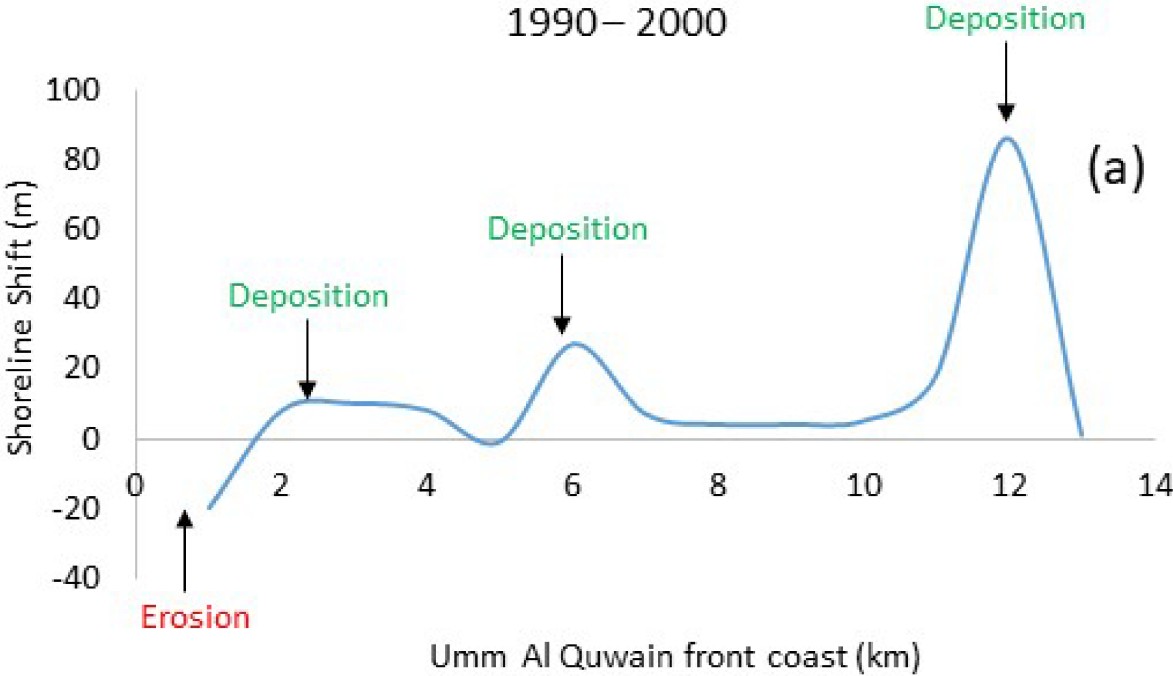

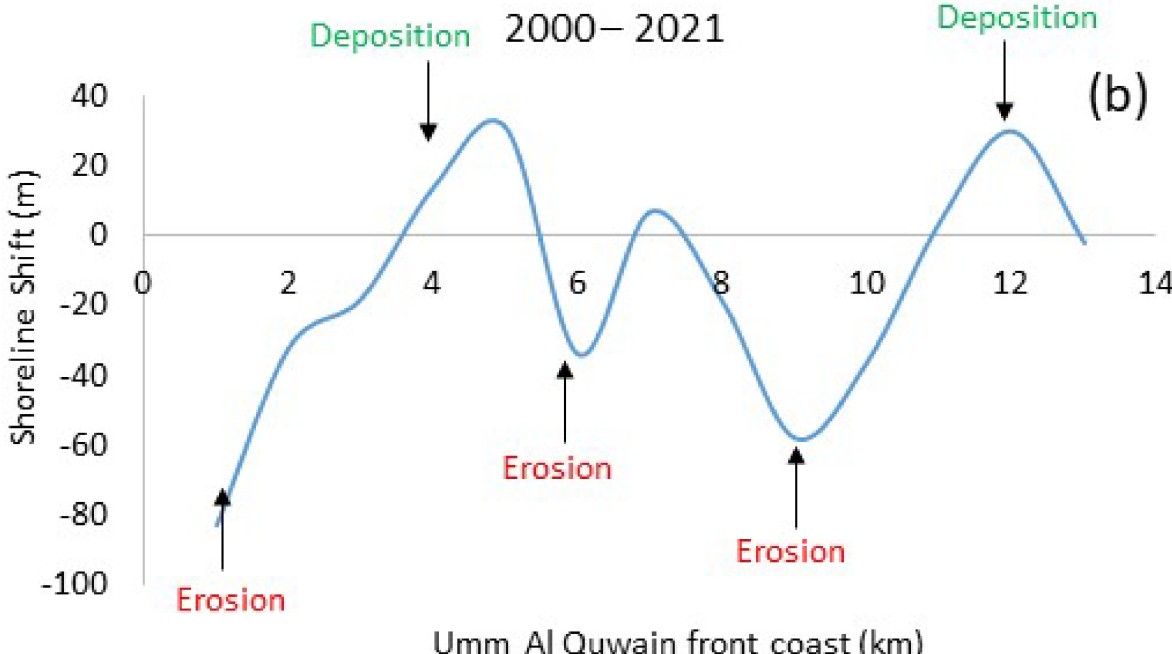

**Figure 10.** Shoreline shift changes along the 14 km long UAQ front coast from (**a**) 1990–2000 and (**b**) 2000–2021. The positive coastline shift caused by accumulation is greater in the upper panel (**a**), whereas the negative coastline shift caused by erosion is greater in the lower panel (**b**).

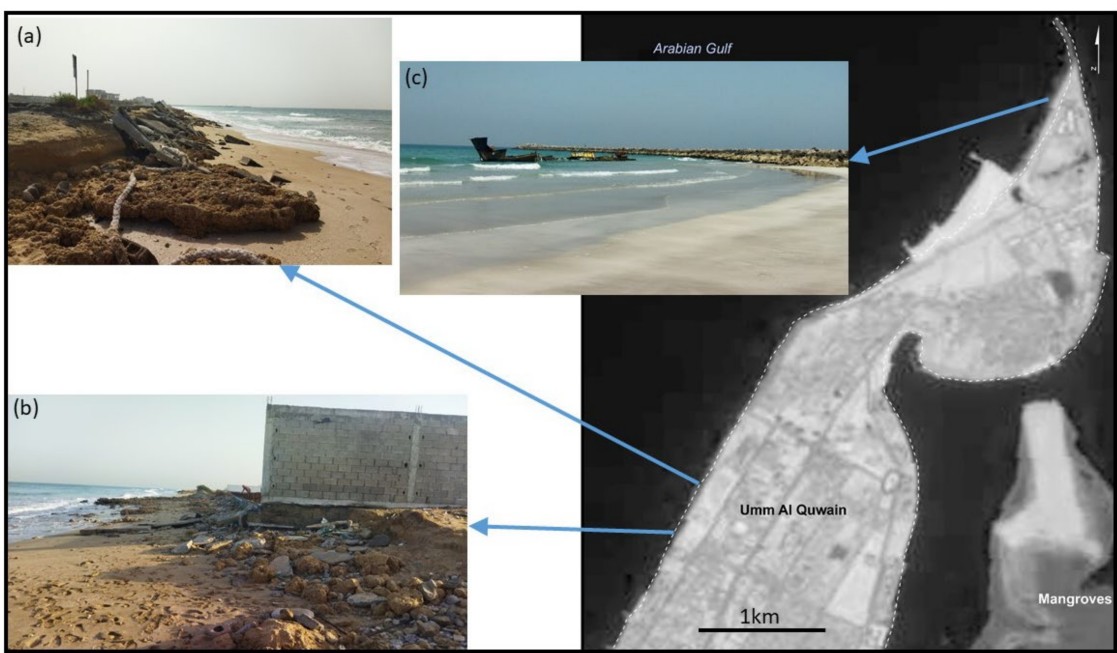

**Figure 11.** Landsat 7 Band 5 image dated 25th November 2021 showing the northern part of the Umm Al Quwain front coast. The dashed line in white denotes the shoreline location in 1990. Insets (**a**,**b**) are pictures taken in the field displaying the nature of coastal retreat along the coast and (**c**) is a shipwreck placed in order to reduce the erosion.

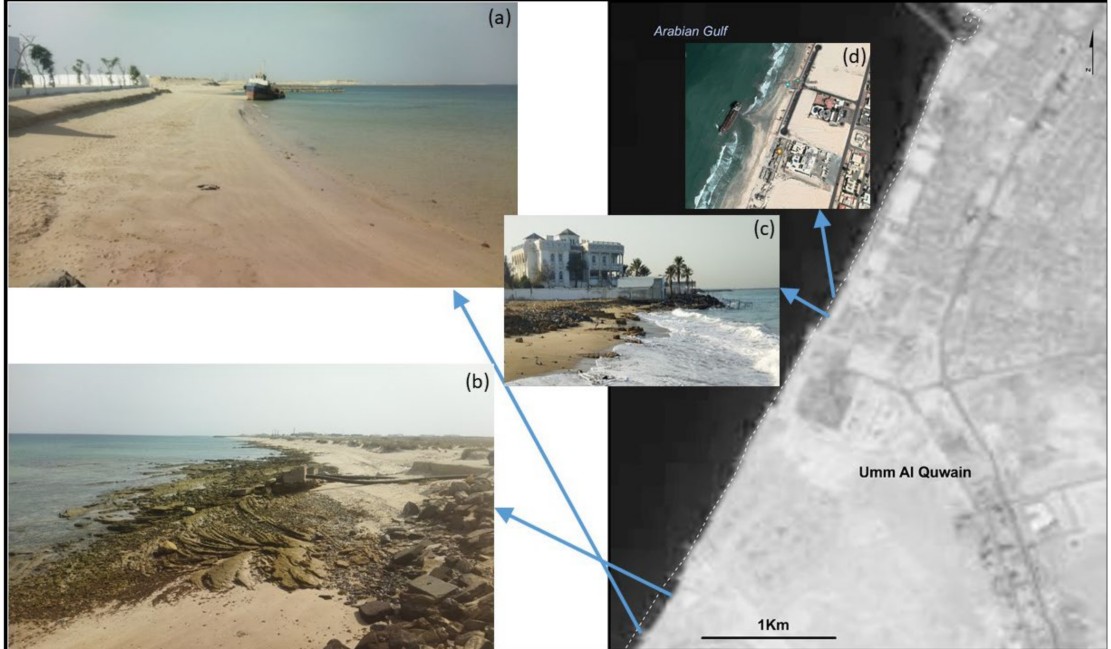

**Figure 12.** Landsat 7 Band 5 image dated 25th November 2021 showing the central part of the Umm Al Quwain front coast. The dashed line in white denotes the shoreline position in 1990. Insets (**b**,**c**) are pictures taken in the field displaying the nature of coastal retreat along the coast and (**a**,**d**) are shipwrecks placed to stop the erosion along the coast.

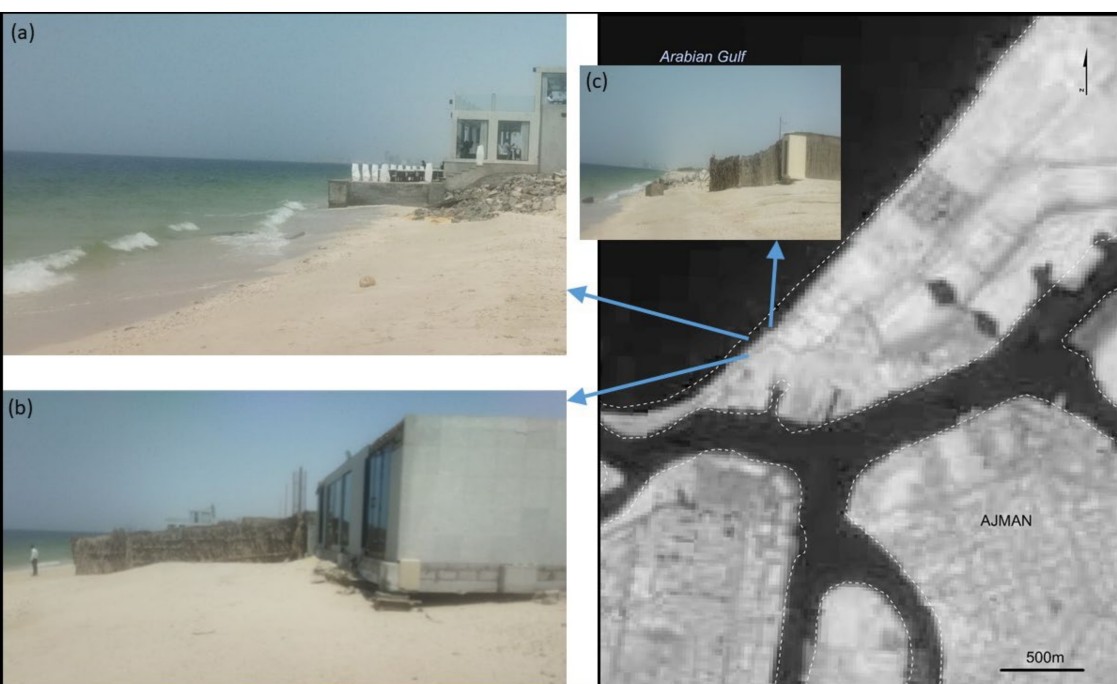

**Figure 13.** Landsat 7 Band 5 image dated 25th November 2021 showing the northern part of Ajman front coast. The dashed line in white denotes the shoreline position in 1990. Insets (**a**–**c**) are pictures taken in the field displaying the nature of coastal retreat along the coast.

Remedies to Save UAQ and Ajman Coast

As the coast along UAQ and Ajman are experiencing widespread erosion affecting important assets such as residential villas, hotels, parks, lamp posts for lighting at night, etc., along the beach, the present situation has come to a point that it is difficult to go back and remove all the encroachments and leave the beach in its pristine condition. Therefore, it is very important during the design and construction phases of ARIs for different models to be used to augment the sustainability and mitigate the negative environmental impacts.

1. Groins/Breakwaters Are of No Use

A groin is a rigid hydraulic concrete structure perpendicular to the shoreline that interrupts water flow and restricts sediment movement. The construction of groins to save the eroding beach would be ineffective because groins built perpendicular to the beach would become new headlands, causing further beach erosion. Although, if a series of these structures were constructed at closer intervals, the seasonal shift in the path of longshore drift would cause the beach crenulation to change seasonally, as observed by [136,137]. Depending on the longshore drift, the beach becomes more dynamic as erosion occurs on either side of the groin. Eventually, the beach's stability is compromised when groins are not present.

2. Seawalls

Seawalls are a type of shore protection structure built where the sea and associated coastal processes have a direct impact on the coastline's landforms. A seawall's purpose is to protect areas where people conduct habitation and recreational activities from the effects of waves and tides. However, seawalls may not be effective in this region. The collapsed structure along the coast of UAQ should serve as a cautionary tale (Figures 13c and 11a,b). As stated previously, seawalls are known to exacerbate the erosion problem, as is the case in Kerala, India [138] and Mumbai, India [139]. As seen from the field in UAQ, a collapsed compound wall is the primary issue with seawalls because waves directly impact them, particularly during the high spring tide, and are reflected into the water, dragging the beach material (Figure 11b). In addition, seawalls avert sand from dunes and cliffs behind

them from reaching the receding beach that would otherwise nourish the beaches. The consequence is the loss of beach and the eventual collapse of the seawalls.

3.    Offshore Barriers/Detached Breakwater—Probable Solution

The purpose of offshore breakwaters is to safeguard a coast or coastal activities from strong wave action. Generally speaking, an isolated breakwater is a shoreline-parallel structure situated within or near the surf zone. Europe and the United States have utilized detached offshore breakwaters as a form of coastal protection since the mid-1960s. Since the 1970s, they have flourished in Japan, with more than 900 units built by 1974 [140] and more than 4800 by 1989 [141] In addition to Japan, they are built in Singapore, Israel, Egypt, France, Spain, and the United States [142,143]. Offshore breakwaters are typically shore-parallel, but are sometimes placed tangentially to provide greater protection from a specific direction [144]. In the majority of instances, these offshore breakwaters are used to stabilize and retain beach material by extending the time between replenishments. In other instances, they have been employed to create or expand the beach width for recreational purposes [145–147].

Construction of offshore barriers appears to be a viable option for protecting the beaches in UAQ and Ajman under current conditions, and furthermore, these coastal areas within the Arabian Gulf fall under a microtidal environment as mentioned by Subraelu [105] which makes the process easier. These parallel concrete structures are typically constructed in the offshore region. Generally, tetrapod blocks are placed on the seafloor to a height determined by the local high tide level. Each of these breakwaters would be a few meters in length and parallel to the shore, with a gap to allow seawater to flow towards the shore. Consequently, a number of these barriers aligned in a line based on local requirements would effectively prevent direct wave action from reaching the shore. Landward of these offshore breakwaters, a tranquil lagoon-like environment is created. In this region, deposition is promoted as the material carried by the waves is carried by eddy currents into the relatively calm area behind the barriers, where it settles [148]. Therefore, the beach would be tranquil and safe for beachgoers. This is evidenced by the artificial reefs created by shipwrecks off the coasts of UAQ and Ajman (Figure 12a,d), which contributed to the formation of beautiful salient or tombolo beaches. Salient beaches are a protrusion in the shoreline on the leeward side of the breakwater, whereas tombolos are beaches that extend to and connect with the breakwater [149]. The only issue with these artificial reefs, such as shipwrecks, is that they detract from the beauty of the area, but they do help prevent beach erosion. These offshore breakwaters can be built parallel to beaches that are always below the level of low tide in that region. As they are always invisible to beachgoers, they effectively reduce wave assault while preserving the area's natural beauty. The planners and administrators of the governments of UAQ and Ajman may consider these options to preserve the beaches for which the cities are renowned for their beauty.

## 6. Conclusions

This study describes the United Arab Emirates (UAE) as a nation, along with the level by which the people are utilizing and altering coastline morphology through the application of exceptional geoengineering techniques. Creating land in water is undeniably a prevalent worldwide phenomenon. Clearly, population pressure is one of the driving forces behind the creation of new artificial land over the ocean [102], although it is not the only one. On an individual emirate basis, it is necessary to conduct additional research into the differences in land reclamation's drivers, processes, and potential effects. The study of coastal expansion exemplified by all UAE coastal emirates provides unmistakable scientific proof of the severity and rate of building far beyond the land and poses a number of pertinent questions to stimulate further investigation.

(1)    How do these structures react in the perspective of sea level rise, climate change, land subsidence, earthquakes, and shoreline erosion along adjacent coastlines caused by sediment depletion?

(2)  What are the boosters of this kind of development, and in what ways do they vary between the various emirates of the United Arab Emirates?

(3)  What materials are used to build these artificial lands in water and where is the source?

(4)  What will be the likely geomorphic as well as other environmental concerns in the area of the source material and at the project site, and how can they be minimized and sorted in the most environmentally friendly manner?

In addition to the preceding, the findings of this study highlight the intensity of the land reclamation process in the United Arab Emirates, of which a majority of coastal emirates have already embraced it. The geomorphic categorization of structures constructed as an outcome of geoengineering just on the coast displayed here allows for additional research into the reasons for constructing such distinguishable buildings. Land reclamation in Dubai, for instance, differs from that in Abu Dhabi, RAK, and Fujairah. A thorough assessment of this necessitates a multidisciplinary approach that includes both geoengineering and physical oceanography perspectives in order to understand what is driving demand, what variables influence the specific design of land extension, and precisely what its environmental effects are and how they can be minimized.

Furthermore, the United Arab Emirates is vulnerable to geological dangers such as earthquakes, storms, and tidal surges. Iran and its environs are subject to moderate to strong seismic activity as a result of the Arabian Plate's ongoing subduction under the Eurasian Plate. In the previous 10 months, the Arabian Gulf coast pertaining to southern Iran has had a number of earthquakes of high magnitude, with vibrations always felt in Dubai, Sharjah, and other northern emirates. Due to these tremors, the Arabian Gulf's surface and underwater currents represent a serious erosion, slope failure, and liquefaction hazard to the reclaimed islands. To maintain these developments, further stabilizing and continuous monitoring efforts should be needed. Additionally, the conspicuous coastal erosion in Ajman and UAQ due to the deprival of sediments because of the construction of artificially reclaimed islands in the upfront coast needs to be monitored and controlled using offshore breakwaters.

To conclude that both environmental and human-induced drivers are at work in tandem during the fast increase in artificial reclaimed coastal lands in the UAE, additional research should focus on determining the relationship between these crucial drivers. Since remote sensing data are readily accessible, it has become necessary to verify, evaluate, and visualize the changing trends in marine and coastal environments. Contrary to the background of climate change, eustatic sea level rise, and other environmental challenges and changes in economic conditions, it would be highly motivating to investigate the geomorphic capacity of these evolving artificially reclaimed offshore sites in terms of their long-term sustainability. In addition, baseline studies are mandatory, as they play a crucial role in deciding an appropriate approach for a specific emirate. In addition, monitoring the effectiveness of particular construction strategies to mitigate the potential environmental effects of ARIs in the UAE should be the subject of additional research.

**Author Contributions:** Conceptualization, P.S.; methodology, P.S. and A.S.; software, P.S.; validation, P.S., A.S. and A.A.E.; formal analysis, P.S.; investigation, P.S.; resources, P.S; data curation, P.S.; writing—original draft preparation, P.S.; writing—review and editing, P.S., K.N.R. and M.M.Y.; visualization, P.S.; supervision, P.S. and M.S.; project administration, P.S. All authors have read and agreed to the published version of the manuscript.

**Funding:** The National Water and Energy Center of the United Arab Emirates University has supported this research. The authors would like to thank the journal's editorial board and reviewers for their professional assistance.

**Conflicts of Interest:** The authors declare no conflict of interest.

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
