# Peer review of "Land in Water: The Study of Land Reclamation and Artificial Islands Formation in the UAE Coastal Zone: A Remote Sensing and GIS Perspective"

_land, doi:10.3390/land11112024_

Round 1

Reviewer 1 Report

The topic of the article is interesting, and appropriate for publication in LAND. Although the intent of the work is good, the manuscript needs a revamping to elevate its quality. This work not well balanced.

My comments for this article are as follows:

The article is very long, confusing and very lacking in precision.

The article must be revised and rewritten succinctly and objectively.

The papers should objectively indicate the contribution of the work to the field and the progress of the research.

In the abstract is referred “…coast experienced a notable shoreline retreat with a net erosion area of 300 m2 and annual rate of 30my−1 over the past 21 years (2000-2021). How did the authors arrive at this result?

Discussion needs to be revised. In this chapter there is text that should be presented in other chapters. For example:

-  the information referred to in lines 341 to 353 about population should not be in the discussion chapter;

-     what is referred to in lines 361 to 363 (Discussion chapter) “Initially, all sentinel–II images for the year 2022 were precisely georeferenced and 361 smoothly combined into a single mosaic image of the Dubai emirate. The image mosaic is subsequently subset with the aid of Dubai's boundary layer.”, is part of the methodology;

-          LULC classes assessed for Dubai (Figure 7) should be in chapter 4. Results.

In Line 626 -628 refers to the Revised Universal Soil Loss Equation which can evaluate the annual beach erosion rates. It should be noted that RUSLE is for estimating soil loss in watersheds.

The discussion chapter presents information that is not supported by the results.

The conclusions are generic and do not reflect the article.

Author Response

Response to Reviewer 1 comments:

1) In the abstract is referred “…coast experienced a notable shoreline retreat with a net erosion area of 300 m2 and annual rate of 30my−1 over the past 21 years (2000-2021). How did the authors arrive at this result?

Response 1: First digitizing the shorelines from the respective dated images (2000 and 2021) and by comparing the two shorelines and merging them in Arc GIS using ArcTools analysis techniques, and then taking the areas under erosion and deposition, once arrived here, then the rate is calculated.

Discussion needs to be revised. In this chapter there is text that should be presented in other chapters. For example:

-      2) The information referred to in lines 341 to 353 about population should not be in the discussion chapter;

Response 2: As per the suggestion, we changed it accordingly.

-       3) what is referred to in lines 361 to 363 (Discussion chapter) “Initially, all sentinel–II images for the year 2022 were precisely georeferenced and 361 smoothly combined into a single mosaic image of the Dubai emirate. The image mosaic is subsequently subset with the aid of Dubai's boundary layer.”, is part of the methodology;

Response 3; Added to the methodology section.

-       4) LULC classes assessed for Dubai (Figure 7) should be in chapter 4. Results.

      Response 4: Added to the Results section.

  5) In Line 626 -628 refers to the Revised Universal Soil Loss Equation which can evaluate the annual beach erosion rates. It should be noted that RUSLE is for estimating soil loss in watersheds.

Response 5: Got confused about the model is used for estimating coastal erosion! Removed it.

  6) The discussion chapter presents information that is not supported by the results.

 Response 6: Modified the Results chapter accordingly.

  7) The conclusions are generic and do not reflect the article

    Response 7: Changed as per your advice, included aspects on the threats of earthquakes and shoreline         

    erosion and ways to control and monitor it periodically.

Reviewer 2 Report

This article appears to be well organized and structured and uses an appropriate methodology to assess the increase in landward expansion towards the sea to consider the possible causes of extensive construction in UAE and the environmental impact of this land expansion. I didn't notice any significant errors in the text, figures, or tables. However, some minor mistakes need to be corrected: when the results are presented in km2, it may not be necessary to enter parts of km because it creates confusion as in lines 374 is 226km2, 375 is 76.68 km2 and 377 is 1.189 km2. It is not clear what sizes are involved. Also, in line 399 it says that 60,489 km2 was added to the list of reclaimed territories along the coast of Dubai. Again, line 407 is 67,349.9 km2.  Figure 10 shows the displacement of the coastline in meters, and in the text, in lines 570 to 573, it is written about m2. Changes in the area of 100 m2 are difficult to see on Landsat images. In line 670 it is not clear what is several meters long.

Author Response

Response to Reviewer 2 comments:

1) This article appears to be well organized and structured and uses an appropriate methodology to assess the increase in landward expansion towards the sea to consider the possible causes of extensive construction in UAE and the environmental impact of this land expansion. I didn't notice any significant errors in the text, figures, or tables. However, some minor mistakes need to be corrected: when the results are presented in km2, it may not be necessary to enter parts of km because it creates confusion as in lines 374 is 226km2, 375 is 76.68 km2 and 377 is 1.189 km2. It is not clear what sizes are involved. Also, in line 399 it says that 60,489 km2 was added to the list of reclaimed territories along the coast of Dubai. Again, line 407 is 67,349 km2.  Figure 10 shows the displacement of the coastline in meters, and in the text, in lines 570 to 573, it is written about m2. Changes in the area of 100 m2 are difficult to see on Landsat images. In line 670 it is not clear what is several meters long.

Response 1: I just wanted to make it clear that the results are in km2 (it’s a typological error) means representing an area which is km2 (Square kilometers) in the lines pertaining to line 374 is 226km2, line 375 is 76.68km2 and line 377 is 1.189km2.

Coming to the line 399, it says 60.489 km2 (Square kilometers) was added to the list of reclaimed territories along the coast of Dubai.

In line 407, 67.349 km2 means 67.349 km2 (Square kilometers).

In lines 570-573 it is written about m2 means represents an area in m2 (Square meters).

Changes in the area of 100m2 are difficult to see on Landsat images?

Actually we have digitized shorelines from Landsat images pertaining to different dates like 1990, 2000, 2010 and 2021. After merging the shorelines in ArcGIS software, then we get the areas of deposition and erosion.

Reviewer 3 Report

A general overview:

The paper I reviewed has been completed. English is well-used and well-written. Although there are a few minor grammatical and word errors in the paper, it has a very readable scientific style. Authors need to recheck their papers in order to resolve these errors. I have provided detailed comments below. An overview of recent progress is presented in the paper. Basically, this paper summarizes the current state of knowledge on the topic. Discussing recent research papers, helps the reader gain a deeper understanding of the topic.

Specific comments:

1. The proposed paper is very well structured. In the Introduction section, there is an effort to provide previous studies with similar scientific content, and in some cases in other countries. However, the available literature is very limited. Generally, the literature on the paper is very poor. This is not acceptable for a research paper that deals with Remote sensing. How do you address the research gap and the novelty of your paper, without properly stating the art? Your paper should be read by researchers from all over the world. This drastically limits the reader's audience. Find more references to the classification of different geo-engineered reactions. You may have some literature to add to your paper.

2. At the end of the Discussion section, the authors clearly state that the goal of the research is to establish environmental correlates that can be objectively measured using remote sensing. Are they protective of the possible potential impacts and methods to mitigate the risks of artificial reclamation of land targets?

3. The methodology is generally very interesting, but not well explained, so other researchers could easily repeat it. More details about the models and their results should be added. This methodology, however, is not novel, as it incorporates already known and existing analyses and procedures.

4. The results are well stated and in my opinion, the tables and figures are easily understandable. However, I believe that some changes need to be made. I suggest separating the Results and Discussions. It will be easier for readers to check the papers and find relevant findings.

5. The paper clearly indicates that it is reporting preliminary results on a map of a critical perspective analysis of potential impacts and risks of ARI in the type of shoreline retreat within the various emirates of the UAE. The paper, however, presents a practically viable technique, which I think would be very useful for the development of a more practical approach. Where is a comparison of your results with other approaches? Efforts to develop an empirically justified method of human-health probabilistic risk assessment can be seen in this paper.

Constructive feedback:

When did you collect the data? It is recommended to identify the sampling date's initial and final day and year. Did they grab water samples or did they collect them automatically? Are the samples collected randomly or were they collected after or during precipitation events?

The narrative about the study area does not include relevant information about the specific location of Umm Al Quwain (UAQ) and Ajman. There is no description of social events (e.g. climate and weather (e.g. annual precipitation, evapotranspiration) nor a description of the area (e.g. ) relevant to the interpretation of the contribution of the study areas (e.g. the distribution and accessibility of a probable victim of artificial reclaimed lands construction). It would be appropriate to include a map of all study areas. I believe that weather conditions are the most important factor to consider when building offshore barriers, for instance.

I think, in its present form, with improvements suggested in the previous section, the paper makes an acceptable case for publication. I think this will generate research interest in improving the two components mentioned above.

Briefly summarize:

In this paper, the results of several primary literature papers are combined to create a coherent argument about a subject or a focused description of a field. Abstracts adequately describe the manuscript. The text summarizes the topic concisely. The purpose of the study is clear. A review of the actual literature provides the authors with a rationale for performing the study. However, the methodology needs to be expanded. A thorough description of the methodology is provided. The methods can be replicated by other investigators. The results are logical and feasible. The conclusions of the authors are supported by the results of the research.

Author Response

Response to Reviewer 3 comments:

1) The proposed paper is very well structured. In the Introduction section, there is an effort to provide previous studies with similar scientific content, and in some cases in other countries. However, the available literature is very limited. Generally, the literature on the paper is very poor. This is not acceptable for a research paper that deals with Remote sensing. How do you address the research gap and the novelty of your paper, without properly stating the art? Your paper should be read by researchers from all over the world. This drastically limits the reader's audience. Find more references to the classification of different geo-engineered reactions. You may have some literature to add to your paper.

Response 1: As per your suggestions, we included more references for using remote sensing data [52-54] and also modern geo-engineering technologies [23, 97, 98, 99, 100, and 101].

  1. At the end of the Discussion section, the authors clearly state that the goal of the research is to establish environmental correlates that can be objectively measured using remote sensing. Are they protective of the possible potential impacts and methods to mitigate the risks of artificial reclamation of land targets?

Response 2: There were modern technologies with eco-engineering technologies have been referred to and cited, which will reduce the harmful effects during the construction of Artificial islands.

The artificial islands and the reclaimed lands in Dubai and other emirates are well protected with outer boundaries having breakwaters that will protect them from storm surges and high waves.

  1. The methodology is generally very interesting, but not well explained, so other researchers could easily repeat it. More details about the models and their results should be added. This methodology, however, is not novel, as it incorporates already known and existing analyses and procedures.

Response 3: Included population data as the main criteria for the construction of Artificial reclaimed islands, and also included the image processing steps in the methodology.

  1. The results are well stated and in my opinion, the tables and figures are easily understandable. However, I believe that some changes need to be made. I suggest separating the Results and Discussions. It will be easier for readers to check the papers and find relevant findings.

Response 4: I have accordingly corrected the Results and Discussion section. I have moved the Land use/Land cover details from the Discussion to the results section. Also changed the population information and population graph (Figure 6) to the Results section.

  1. The paper clearly indicates that it is reporting preliminary results on a map of a critical perspective analysis of potential impacts and risks of ARI in the type of shoreline retreat within the various emirates of the UAE. The paper, however, presents a practically viable technique, which I think would be very useful for the development of a more practical approach. Where is a comparison of your results with other approaches? Efforts to develop an empirically justified method of human-health probabilistic risk assessment can be seen in this paper.

Response 5: This paper demonstrates how artificial reclamation of lands might cause conspicuous shoreline changes in the down coast and also UAE’s proximity of the Zagros fold belt region, experienced number of high magnitude earthquakes (more than 6 Mw) in the last few months. Due to these earthquakes, it is imperative to increase the frequency of monitoring the islands base structure and the surrounding areas.

6) When did you collect the data? It is recommended to identify the sampling date's initial and final day and year. Did they grab water samples or did they collect them automatically? Are the samples collected randomly or were they collected after or during precipitation events?

Response 6: In this paper, there is no water samples data taken for the study as we are not studying the water quality parameters.

7) The narrative about the study area does not include relevant information about the specific location of Umm Al Quwain (UAQ) and Ajman. There is no description of social events (e.g. climate and weather (e.g. annual precipitation, evapotranspiration) nor a description of the area (e.g.) relevant to the interpretation of the contribution of the study areas (e.g. the distribution and accessibility of a probable victim of artificial reclaimed lands construction). It would be appropriate to include a map of all study areas. I believe that weather conditions are the most important factor to consider when building offshore barriers, for instance.

Response 7: We have included the relevant information on UAQ and Ajman coastal areas and shoreline lengths in the Study area section. In our study we haven’t taken weather conditions in to consideration, as they are important to consider during the construction of reclaimed of islands. The area within the Arabian Gulf and along the UAE coast comprises of mainly micro tidal environment which makes the process easier.

Round 2

Reviewer 1 Report

With the review prepared by the authors, the article improved.

Reviewer 3 Report

Thank you for proving this comment and reviewing the paper.